# Chemical Characterization and Immunomodulatory Activity of Fucoidan from *Sargassum hemiphyllum*

**DOI:** 10.3390/md21010018

**Published:** 2022-12-26

**Authors:** Rui Li, Qing-Ling Zhou, Shu-Tong Chen, Min-Rui Tai, Hong-Ying Cai, Rui Ding, Xiao-Fei Liu, Jian-Ping Chen, Lian-Xiang Luo, Sai-Yi Zhong

**Affiliations:** 1College of Food Science and Technology, Guangdong Ocean University, Guangdong Province Engineering Laboratory for Marine Biological Products, Guangdong Provincial Engineering Technology Research Center of Marine Food, Guangdong Provincial Key Laboratory of Aquatic Product Processing and Safety, Guangdong Provincial Science and Technology Innovation Center for Subtropical Fruit and Vegetable Processing, Zhanjiang 524008, China; 2Collaborative Innovation Center of Seafood Deep Processing, Dalian Polytechnic University, Dalian 116034, China; 3The Marine Biomedical Research Institute, Guangdong Medical University, the Marine Biomedical Research Institute of Guangdong Zhanjiang, Zhanjiang 524023, China

**Keywords:** *Sargassum hemiphyllum*, fucoidan, chemical characterization, immune-boosting activity, functional food

## Abstract

Fucoidan is a sulfated algal polyanionic polysaccharide that possesses many biological activities. In this paper, a fucoidan (SHF) polysaccharide was extracted from *Sargassum hemiphyllum* collected in the South China Sea. The SHF, with a molecular weight of 1166.48 kDa (44.06%, *w*/*w*), consisted of glucose (32.68%, *w*/*w*), galactose (24.81%, *w*/*w*), fucose (20.75%, *w*/*w*), xylose (6.98%, *w*/*w*), mannose (2.76%, *w*/*w*), other neutral monosaccharides, and three uronic acids, including glucuronic acid (5.39%, *w*/*w*), mannuronic acid (1.76%, *w*/*w*), and guronuronic acid (1.76%, *w*/*w*). The SHF exhibited excellent immunostimulatory activity. An immunostimulating assay showed that SHF could significantly increase NO secretion in macrophage RAW 264.7 cells via upregulation of cyclooxygenase-2 (COX-2) and inducible nitric oxide synthase (iNOS) levels based on both gene expression and protein abundance. These results suggest that SHF isolated from *Sargassum hemiphyllum* has great potential to act as a health-boosting ingredient in the pharmaceutical and functional-food fields.

## 1. Introduction

More attention has been paid recently to the innate immune system due to an increase in chronic diseases and undesirable lifestyles, especially under the prevalence of COVID-19 [1]. These unhealthy factors may lead to hypoimmunity of the human body: a status of permanent or temporary immune disorders that makes living organisms more susceptible to causative agents, resulting in injury of the immune system [2]. With immunoreaction cumulatively recognized as a vital constituent of favorable therapeutic outcomes, immunity regulation has arisen as a potential approach to conquer this challenge [3]. In order to prevent and treat immunosuppressive diseases, seeking and developing natural functional-food phytochemicals is one of the most attractive and effective methods.

Macrophages have been extensively employed in immunology studies, since they are able to trigger inherent immunoreactions and produce the appropriate immune responses in answer to immunological illness, microbial infection, and cancer [4]. The immune system is closely connected to the appearance and progress of many diseases, including inflammation [5], cancer [6], and diabetes [7]. Moreover, production of proinflammatory factors is another aspect of the immune system’s response to inflammation, injury, or pathogens [8]. Nitric oxide (NO) has been shown to act as a primary mediator of macrophages, which are vital for immune-system resistance to pathogen aggression [9]. Inducible nitric oxide synthase (iNOS) has been shown to be a key downstream intervenor of inflammation in a variety of cell types [10], and cyclooxygenase-2 (COX-2) is a crucial inflammatory mediator [11]. Studies have shown that production of iNOS and COX-2 is remarkably increased in LPS-induced RAW 264.7 cells [12]. 

There are 250 genera and 1500 species of brown seaweed (Phaeophyta) in the marine area across the world [13]. Sargassum belongs to the brown seaweed class and includes about 400 species [14] in the world. There are at least 60 species of Sargassum throughout China. Most species grow in the region of the South China Sea [15] and contain many bioactive components, such as polysaccharides, polyphenols, proteins, peptides, vitamins, and minerals [16]. Sargassum sulfated polysaccharides, known as fucoidans, exhibit diverse biological activities with potential health benefits, such as immune-system boosting, anti-inflammation, and antiviral activity [17]. 

Much attention has been paid to the immunomodulatory activities of polysaccharides, and a range of research has demonstrated that polysaccharides could restrain tumor metabolic processes in vivo depending on immune regulation [18]. Research has also shown that sulfated polysaccharides derived from brown seaweed have strong immunomodulatory effects in vivo and in vitro, demonstrating exclusion of cancer cells and virus-infected cells [19]. Moreover, certain polysaccharides have exhibited effective immune-system stimulation via direct or indirect interactions, as well as enhancement of particular mechanisms of host responses [20]. 

The biological activity levels of polysaccharides are intimately related to their structures, including α/β-configuration, monosaccharide composition, branching degree, glycosidic linkage, and side-chain length, as well as harvest season and algae location [21]. Hence, it is essential to study the structure and activities of seaweed polysaccharides from different areas to exploit more food and pharmaceutical resources. However, to date, there are only a few studies about the antioxidant activity, the immune-stimulating activity [22], and the anti-inflammation activity [23] of fucoidan from *Sargassum hemiphyllum*. Therefore, more research of fucoidan from *Sargassum hemiphyllum* is necessary.

In this work, we aimed to investigate the chemical properties and immunomodulatory activities of fucoidan extracted from *Sargassum hemiphyllum* (SHF). Fucoidan was extracted from the brown algae *Sargassum hemiphyllum* collected on the coast of the South China Sea in Guangdong, China. The SHF chemical structure was analyzed through Fourier transform infrared spectroscopy (FTIR), an ion chromatograph, high-performance gel-permeation chromatography, etc. The immunoregulatory activities of fucoidans on macrophage RAW 264.7 cells were determined via testing of NO production and COX-2 and iNOS expression at both the protein and the gene level. We found that SHF had excellent immunostimulatory activity. We believe this work provides a theoretical foundation for further development of SHF as a functional-food ingredient or a nutraceutical supplement to boost the immune system of the human body. 

## 2. Results and Discussion

### 2.1. Yields and Chemical Analysis of Fucoidan Obtained from Sargassum hemiphyllum

Fucoidan was extracted from the brown seaweed species *Sargassum hemiphyllum* (SHF), with a yield rate (based on the raw material) of 2.72% ± 0.18% and an extraction rate (based on the total polysaccharide content of the raw material) of 16.37 ± 1.96% (Table 1). Tabarsa et al. achieved a higher crude fucoidan yield rate of 6.5% from *Nizamuddinia zanardinii* [24]. The yield rate obtained by Liu et al. varied from 3.94% to 11.24% using different extraction methods [25]. 

The total polysaccharide and total protein contents of the SHF were 75.35 ± 1.46% and 2.66 ± 0.67%, respectively (Table 1). Liu et al. found that the total polysaccharide content of *Sargassum fusiforme* was between 63.53% and 72.90% and the total protein ranged from 0.92% to 4.70% [25]. The total polyphenol content of the SHF was 0.49 ± 0.01%. Thahira et al. [26] found that the total phenolic content of extracts obtained from a red algae, *Kappaphycus alvarezii,* and a brown algae, *Sargassum tenerrimum* were 0.045% and 0.031%, respectively, which were about 10 times lower than the polyphenol content of SHF in our experiment. Sulfate groups result in polysaccharides being negatively charged, which is important to the biological properties of fucoidans [27]. Our results demonstrate that the sulfate content of the SHF was 44.11 ± 0.01% (Table.1). Our previous results showed that the sulfate groups of commercial fucoidans from different brown-algae sources ranged from 38.50% to 57.54% [28].

### 2.2. Molecular Properties

Molecular weight properties are important to the overall fucoidan polysaccharide structure because molecular weight distribution determines the polymeric level of a molecular structure [29]. Table 2 provides information about the molecular properties of the SHF, including molecular weight averages (Mw), molecular weight number averages (Mn), and the molecular weight of each highest peak (Mp). 

Mw refers to the molecular size of the sample, which is more affected by higher molecular weight chains [30]. Table 2 demonstrates that there were six portions for the Mw of SHF: 3374.86 kDa (7.68%), 1166.48 kDa (44.06%), 152.37 kDa (14.50%), 121.93 kDa (13.43%), 44.94 kDa (5.91%), and 22.40 kDa (14.42%). Mn represents the statistical average molecular weight of a polymer, which is affected more by lower molecular weight chains [31]. Mp represents the molecular weight distribution mode [30]. The Mn and Mp of SHF were 628.33 kDa (44.06%) and 773.35 kDa (44.06%), respectively, both were the maximum components of the extract. These data show that SHF is highly heterogeneous and has a broad molecular weight distribution.

### 2.3. Monosaccharide Composition

In order to identify the basic physicochemical information of polysaccharides, it is necessary to analyze their monosaccharide compositions [25]. Figure 1 shows the monosaccharide chromatogram of SHF analyzed with ion chromatography, which is the relatively most accurate approach for polysaccharide-structure identification, as it has been supported by standards. According to the data in Table 3, we could conclude that SHF was mainly composed of glucose (32.68%); galactose (24.81%); fucose (20.75%); xylose (6.98%); mannose (2.76%); other neutral monosaccharides; and three uronic acids, including glucuronic acid (5.39%), mannuronic acid (1.76%), and guronuronic acid (1.76%); indicating that SHF is an acidic polysaccharide. 

### 2.4. FT-IR Analysis

The SHF infrared spectrum within the range of 4000–500 cm^−1^ is shown in Figure 2. The major bands of the SHF at 3444 cm^−1^ were due to the SHF’s symmetrical and asymmetrical O-H and H_2_O stretching vibrations [32]. Another weak signal was recorded at 2938 cm^−1^, indicating the existence of C-H stretching vibrations of the pyranoid ring and C6 groups of fucose [33]. The peak near 1640 cm^−1^ was attributed to asymmetric stretching vibrations of carboxylate (O-C-O), which probably correlated with the uronic acids that are present in fucoidan [34]. The band at 1425 cm^−1^ indicated the existence of asymmetrical bending C-H vibrations for glucose, fucose, xylose, and mannose [33]. The band at 1253 cm^−1^ indicated the presence of sulfate-group (S=O) stretching, which resulted in fucoidan branching [35]; this is also the typical band confirmed in fucoidan. Meanwhile, the band at 1136 cm^−1^ was due to the stretching vibrations of the glycosidic C-O group of the fucoidan [35]. A strong signal at near 1037 cm^−1^ was a result of hemiacetal stretching [32]. The weak signals near 968 cm^−1^ were attributed to the asymmetrical stretching vibration of a C-O-S bond [36]. The bands at 831cm^−1^ were assigned to C-O-S bending of the sulfate substitutional group at the axial C-4 position [37]. Kim et al. found similar spectra in fucoidan extracted from the brown seaweed *Undaria pinnatifida* in the range of 820–850 cm^−1^ [37]. A weak peak of the SHF, at 575 cm^-1^, was assigned to O=S=O bending vibration [38]. 

Generally, The FT-IR peaks of the SHF were similar to those of the commercial fucoidans detailed in our previous research [28], and the existence of sulfate groups and a methyl group (from fucose) were the primary characteristics of the SHF, suggesting that the extracts were fucoidans with high purity and composed of similar biofunctional groups to those of commercial fucoidans. Although the spectra were similar to those of commercial fucoidans, some modifications of minor peaks were observed, possibly because of the change in fucoidan chemical structure due to the seaweed’s maturity, species, and geographical region. 

### 2.5. Cellular Nitric Oxide Production 

Nitric oxide production is a characteristic feature of macrophage activation, which protects the body against fierce, growing cancer cells and invasive pathogens. Measurement of NO secretion in a culture medium is one of the most credible ways to assess classical macrophage activation [39]. To investigate whether SHF activates macrophages, mouse macrophage RAW 264.7 cells were cultured with SHF (25, 50, and 100 μg/mL). The NO production thereof was measured and compared with that produced by three commercial fucoidans (Figure 3). For the three commercial fucoidans (Figure 3A–C), none of them increased NO production, in fact they decreased NO production compared to the control group, while a dose-dependent increase trend in NO production was measured in the culture medium of RAW 264.7 cells activated with the SHF among tested concentrations of 25 to 100 μg/mL (Figure 3A–C). Specifically, NO secretion from cells treated with 25 μg/mL SHF had no significant difference from that of the control group. However, higher concentrations, i.e., 50 and 100 μg/mL of SHF, significantly upregulated (*p* < 0.01 and *p* < 0.001, respectively) the release of NO in RAW 264.7 cells compared with the control group (Figure 3D), although there was no significant difference between the 50 μg/mL and 100 μg/mL groups. Therefore, these results indicate that SHF has the ability to promote NO release and macrophages can be activated to participate in immune responses and play an immunomodulatory role.

The functional properties of fucoidans are known to connect to their branching degrees, monosaccharide types, molecular weights, and glycosidic bonds [40]. Sulfate content and molecular weight may be responsible for cell immune responses [41]. Some studies have demonstrated that lower-molecular-weight polysaccharides have better stimulating effects on the innate immune system [42]. Similarly, acidic polysaccharide fractions from *Spirulina platensis* with a lower molecular weight have shown higher immunostimulatory activity [43]; however, they also exhibited significant NO production and cytokines at 500 and 2000 µg/mL, which are much higher than those of the SHF concentration (50–100 μg/mL) used in this study. Although the SHF had a large Mw, it still had good immune-boosting activity in the content range of 50–100 µg/mL. Choi et al. indicated that the existence of acetyl and sulfate groups has a primary effect on fucoidan immunomodulatory activity [44]. In regard to immune regulation, the structure–function relationship of fucoidan is still controversial. Further study of the immunoactivity of the degraded polysaccharide of *Sargassum hemiphyllum* is necessary.

### 2.6. SHF Activates mRNA Expression of COX-2 and iNOS in Macrophages

The common mechanisms of immunomodulatory activities correlate with various macrophage-derived functional factors. iNOS has been shown to be a necessary controller that is responsible for a great amount of NO integration in macrophages, and both COX-2 and iNOS are proinflammation mediators [45]. Therefore, to identify macrophage activation via fucoidans, we chose to research the effects of SHF on COX-2 and iNOS mRNA expression in macrophages, using a *q*RT-PCR analysis. The results thereof indicated that there was no obvious increase in COX-2 mRNA expression when the SHF concentration was at 25 μg/mL, and the mRNA expression of the COX-2 was significantly upregulated by the SHF at 50 μg/mL (*p* < 0.001) and 100 μg/mL (*p* < 0.0001), respectively; however, there was no significant difference between these two groups (*p* > 0.05) (Figure 4A). The expression of the iNOS mRNA was similar to that of the COX-2 (Figure 4B). These results were consistent with the NO production, indicating that SHF may increase target gene-expression levels to promote NO secretion and further sustaining the conclusion that macrophages can be activated with SHF at concentrations of 50 and100 μg/mL. Additionally, SHF increases NO release via upregulation of COX-2 and iNOS mRNA expression.

### 2.7. Immunofluorescence

Immunofluorescence was employed to further determine whether SHF increased the protein expressions of both COX-2 and iNOS. In the SHF-treated group, the blue and green fluorescences of the COX-2 (Figure 5A) and iNOS (Figure 5B) proteins were brighter and had higher fluorescence intensity, with an increase in SHF concentration, demonstrating that the SHF promoted the protein expressions of COX-2 and iNOS in the RAW 264.7 cells. This result was consistent with the NO secretion of the RAW 264.7 cells. 

### 2.8. Western Blot

To further elucidate the activation of the COX-2 and iNOS proteins of the RAW 264.7 cells, a Western blot analysis was used to evaluate the protein expressions of the COX-2 and the iNOS (Figure 6), and the images thereof were analyzed using ImageJ. As shown in Figure 6, the levels of COX-2 and iNOS protein expression increased in the SHF-treated cells after stimulation. All three tested groups (25, 50, and 100 µg/mL) had significantly higher levels of COX-2 and iNOS compared to the control group (*p* < 0.0001). Additionally, the 50 µg/mL and 100 µg/mL groups had significantly higher levels of these two proteins compared to the 25 µg/mL group, although there were no obvious differences between these two higher concentrations. We could see that at a protein level, SHF at concentrations of 25–100 µg/mL could boost the immune system.

Some studies have correlated anti-inflammatory activity with immunomodulatory effects [41]. This is supported by the work of Kang et al, who found that the fucoidan from *Ecklonia cava* inhibited NO production via inhibition of iNOS and COX-2 in LPS-stimulated RAW 264.7 cells [46]; Song et al. also found that *Undaria pinnatifida* sporophyll extract suppressed LPS-induced iNOS and COX-2 expression, indicating an anti-inflammatory effect [47]. Some other research showed that fucoidan had anti-inflammatory activity, and Shikov et al. found that fucoidan extracted from *Fucus vesiculosus* exhibited a relative higher COX-2 inhibitory activity with an IC_50_ value of 4.3 µg/mL, indicating that it had a good anti-inflammatory property [48]. 

The expressions of several common immunomodulatory regulators are correlated with the productions of a variety of macrophage-derived functional factors. It is generally thought that the primary mechanism through which fucoidan secures cells is activation of the innate immune response. This has great potential for regulation of immunomodulatory signaling. For example, some research has shown that fucoidan regulates p38 mitogen-activated protein kinase (p38MAPK) and NF-κB-dependent signaling to upregulate NO secretions from RAW 264.7 cells [49]. Similarly, low-molecular-weight fucoidan (<10 kDa) prepared from New Zealand *Undaria pinnatifida* significantly activated RAW 264.7 cells through regulation of the MAPK and NF-κB signaling pathways [50]. These studies provide a theoretical foundation for the potential of fucoidan as part of the next generation of polysaccharide immune modulators, including those involved in tumor therapy. Therefore, detection and assessment of fucoidan’s immunostimulatory effects is of valuable research interest in the chemistry, biology, and medical fields. 

## 3. Materials and Methods

### 3.1. Reagents

FucU1 fucoidan samples (extracted from Undaria pinnatifida; batch number 572001015; date of receipt, 16 March 2021; expiration date, 10 January 2023) were purchased from Qingdao Bright Moon Seaweed Group Co., Ltd. (Qingdao, China); FucU2 (extracted from Undaria pinnatifida; Lot# SLCH6433; date of receipt, 28 June 2021) and FucF (extracted from *Fucus vesiculosis*; Lot# SLCJ3756; date of receipt, 17 October 2021) were purchased from Sigma-Aldrich (St. Louis, MO, USA). Monosaccharide standards were purchased from BoRui Saccharide Biotech Co. Ltd (Yangzhou, China). Fetal bovine serum (FBS), Dulbecco’s Modified Eagle Medium (DMEM), Trizol reagent, streptomycin, penicillin, and primers were purchased from Sangon Biotech Co., Ltd. (Shanghai, China). A nitric oxide (NO) detection kit, a BeyoECL star kit, and a cytoplasmic and nuclear-protein extraction kit were obtained from Beyotime (Haimen, China). Antibodies for GAPDH, iNOS, and COX-2, and secondary antibodies (HRP-linked antirabbit IgG) were obtained from Abcam (Shanghai, China). Real-time fluorescent quantitative PCR premix (SYBR) and a total RNA isolation kit were purchased from Tiangen Biotechnology Co. (Beijing, China). TransScript® One-Step gDNA Removal and cDNA Synthesis SuperMix were purchased from Transgen Biotechnology Co. (Beijing, China). Other chemicals were analytical-grade and purchased from Xilong Scientific Company (Guangdong, China). Aqueous solutions were prepared with deionized or doubly distilled water. 

### 3.2. Collection of Seaweed

Five batches of *Sargassum hemiphyllum*, each 30 kg, were collected via hand picking from the coastal region of Naozhou Island, Zhanjiang, Guangdong, China in April 2021 and, immediately after being picked in a fresh state, were identified by Professor En-Yi Xie of Guangdong Ocean University, who specializes in the classification of large, economic seaweeds. All algae samples collected were in the breeding period identified by Professor Xie. The collected seaweeds were washed with seawater and fresh water. After washing, the seaweeds were dried under the sunshine until a constant weight was reached, then ground using a blender and sieved using a 100-mesh sieve. The voucher specimens were deposited into a −20 °C fridge at the College of Food Science and Technology, Guangdong Ocean University, Zhanjiang, China. Three batches thereof were used in this study.

### 3.3. Extraction and Purification of Sargassum Fucoidan

For this step, 20.0 g of powder was weighed into a 1 L beaker, and distilled water was added at a ratio of 1:30. A rotor was gently placed into the beaker after stirring using a glass rod, and the beaker was then placed into a thermostatic magnetic stirrer at 80 °C for 3.5 h. Then, a 350 W ultrasonic-assisted extraction was performed for 50 min. The extract solution was centrifuged at 4000 r/min for 10 min, and the precipitation was discarded. After the supernatant was concentrated to one third of its original volume via rotary evaporator, anhydrous ethanol was added to its 30% (*v*/*v*) amount, and the supernatant was centrifuged at 4000 r/min for 10 min again. Anhydrous ethanol was added to the supernatant until its concentration reached 80% (*v*/*v*). The precipitation was washed twice with anhydrous ethanol and acetone, respectively, then dissolved in distilled water to a total volume of 30 mL, and a savage reagent was added in a 6:1 ratio (*v*/*v*) to remove protein. After 20 min of oscillation with the vortex oscillator, the supernatant was placed in a dialysis bag (15000 kDa) and dialyzed in a refrigerator at 4 °C for 24 h, during which the distilled water was replaced 2–3 times. After dialysis, the polysaccharide solution was freeze-dried, and fucoidan was obtained. 

### 3.4. Yields and Chemical Compositions of Fucoidans

The total polysaccharide content in the dry powder and dry extract was determined via the phenol–sulfuric acid colorimetric method. Yield rate and extraction rate were calculated using equation (1) and (2), respectively. Sulfate content was analyzed via the barium chloride–gelatin method, with potassium sulfate used as a standard [51]. Protein content was determined with the Coomassie bright blue method, with bovine serum protein used as a standard, and the absorbance values of the standard solutions and the sample solutions were measured at 595 nm. The total phenolic content was tested using the Folin and Ciocalteu reagent with a gallic-acid standard, according to the method reported previously [52].
(1)$$\mathrm{Yield}\;\mathrm{rate}\;(\%)=\frac{\mathrm{Weight}\;\mathrm{of}\;\mathrm{fucoidan}}{\mathrm{Weight}\;\mathrm{of}\;\mathrm{seaweed}\;\mathrm{powder}}\;\;100\%$$
(2)$$\mathrm{Extraction}\;\mathrm{rate}\;(\%)=\frac{\mathrm{Weight}\;\mathrm{of}\;\mathrm{fucoidan}}{\mathrm{Weight}\;\mathrm{of}\;\mathrm{total}\;\mathrm{polysaccharide}\;\mathrm{content}\;\mathrm{of}\;\mathrm{seaweed}\;\mathrm{powder}}\;\;100\%$$

### 3.5. Determination of Molecular Properties

The molecular weights of the fucoidan samples were tested via high-performance gel-permeation chromatography (Shimadzu, Australia) with a BRT105-104-102 tandem gel column (7.8 × 300 mm i.d.) (Borui Saccharide, Biotech. Co. Ltd.), according to the method of Guan et al. [53] with minor modifications. The fucoidan samples and commercial fucoidans were dissolved in distilled water at a concentration of 5 mg/mL; after centrifugation at 12000 rpm for 10 min, the supernatant was filtered using a 0.22 µm Millipore membrane and transferred into a 1.8 mL sample vial. The samples were eluted with 0.05 mol/L of NaCl through an HPLC system (Shimadzu Corp., Tokyo, Japan) equipped with a refractive index detector (RI-10A). The column temperature was kept at 40 ± 1 °C and the flow rate was 0.6 mL/min. Then, 20 μL of the supernatant was injected into the column and performed for 60 min. Dextran samples with different relative Mw values (5000, 11,600, 23,800, 48,600, 80,900, 148,000, 273,000, 409,800, 667,800) were used as standards for standard curve generation to determine the molecular properties of the polysaccharide.

### 3.6. Monosaccharide Composition Analysis

Monosaccharide composition was determined using an ion chromatograph (ThermoFisher, America) according to the method reported by Li et al. [54], with minor modifications. Specifically, the fucoidan was hydrolyzed completely using trifluoroacetic acid (TFA, 3 mol/L) at 120 °C for 3 h. The hydrolysate was desiccated to remove excess TFA. Then, 5 mL of distilled water was added into the sample and vortexed, and 50 µL of the sample was pipetted into 950 µL of distilled water, then centrifuged at 12000 rpm for 5 min; the supernatant was used for ion chromatographic analysis with a Dionex Carbopac ^TM^PA20 column (3.0 mm × 150 mm) at a flow rate of 0.3 mL/min and a column temperature of 30 °C. The mobile phase was consisted of A, H_2_O; B, 15 mM NaOH; C, 15 mM NaOH and 100 mM NaAc. A 5 µL sample was injected, and the testing process was monitored through an electrochemical detector.

### 3.7. FTIR Characterization

2 mg of fucoidan and 200 mg of KBr were combined and pure KBr powder was used as blank. The samples were sufficiently mixed, compressed into tablets and placed in an FTIR spectrometer (BRUKER TENSOR-2, BRUKER, Germany) for scanning, with a scanning wavelength range of 400–4000 cm^−1^.

### 3.8. Cell line and Cell Culture

The macrophage RAW 264.7 cells were cultured in DMEM that was supplemented with 10% fetal bovine serum, 100 µg/mL of streptomycin, and 100 U/mL of penicillin at 37 °C in a humidified incubator with 5% CO_2_.

### 3.9. Determination of Nitric Oxide Production

Nitric oxide production was tested as described previously, with a small modification [24]. Briefly, 100 μL of RAW264.7 cells (1 × 10^5^ cells/well) were seeded into a 96-well microplate, then 100 μL of either sample fucoidan (10, 25, and 50 μg/mL) or the culture medium (negative control) was added into the wells. The cells were cultured at 37 °C for 24 h in a humidified atmosphere that contained 5% CO_2_; then, the culture medium was separated and mixed with an equal amount of Griess reagent. The microplate was kept at room temperature for 20 min and absorbance was measured at 563 nm with a microplate reader.

### 3.10. Quantitative Real-Time PCR (qRT-PCR Analysis)

Total RNA was extracted using a Trizol assay kit. Briefly, 1 μg of RNA was tested via qRT-PCR using a qPCR master mix kit. PCR amplification was performed via incorporation of SYBR green (Roche). The primers for mouse iNOS, COX-2, and GAPDH were synthesized by Sangon Biotech (Shanghai). The primer sequences were as follows: iNOS-forward primer: CCTCCTCGTTCAGCTCACCT, iNOS-reverse primer: CAATCCACAACTC-GCTCCAA, COX-2-forward primer: TGAGTACCGCAAACGC TT-CT, COX-2-reverse primer: ACGAGGTT-TTTCCACCAGCA, GAPDH-forward primer: GGTGAAGGTCG-GTGTGAACG, and GAPDH-reverse primer: CTCGCTCCTGGAAGATGGTG.

### 3.11. Immunofluorescence Staining

The RAW 264.7 cells (1 × 10^5^ cells/mL) were incubated with DMEM fucoidan samples (25, 50, and 100 µg/mL) for 24 h. The culture medium was discarded, fixed with 4% polyformaldehyde for 10 min and then blocked with 10% BSA for 1 h. The cells were washed three times with PBS (5 min for each washing). Next, the cells were incubated with the primary COX-2 and iNOS antibodies, respectively, at 4 °C overnight, then washed with TBST three times (10 min each time), followed by incubation with a fluorescence-conjugated secondary antibody at room temperature for another 1 h and then washing with PBS three times. Cells were stained with 4′,6-diamidino-2-phenylindole (DAPI) at 1:1000 for 10 min and then removed via washing with PBS three times. Images were obtained with a Cytation 5 cell-imaging multimode reader (Olympus SpinSR10 spinning disk confocal super resolution microscope, Tokyo, Japan).

### 3.12. Western Blot Analysis

The Western blot analysis of the RAW 264.7 cells treated with fucoidan and the control were conducted with the previously reported methods [55]. The 2 × 10^6^ cells were cultured in 6-well plates for Western blots. The cells were then treated with fucoidan at concentrations of 25, 50, and 100 µg/mL for 2 h, respectively, and cultured medium was used as a negative control. Cells were lysed in the lysis buffer (Cell Signaling Technology, Boston, MA, USA) with 20 mM of Tris–HCl, pH 7.5; 2.5 mM of sodium pyrophosphate; 150 mM of NaCl; 1 mM of EGTA; 1 mM of EDTA; 1 mM of Na_3_VO_4_; 1 mM of β-glycerophosphate; 1 μg/mL of leupeptin; and 1% Triton, supplemented with a HAL T protease and phosphatase inhibitor cocktail (ThermoFisher Scientific, Waltham, MA, USA). A total of 30 μg of protein was electrophoresed in each lane on a 4–12% Bis-Tris SDS–PAGE gel. The protein was transferred to nitrocellulose membranes (ThermoFisher, Waltham, MA, USA); then, the transfer system (ThermoFisher Scientific, MA, USA) and iBlot detection were used. Membranes were incubated in 1× TBST (containing 4% milk) at room temperature for 1 h, then membranes were incubated in the primary antibody solutions at 4 °C overnight, including GAPDH, iNOX, and COX-2 (Amersham Life Science, TX, USA). Subsequently, they were further incubated with antirabbit antibodies conjugated to horseradish phosphatase at room temperature for 1 h. Before development with enhanced chemiluminescence (Amersham Life Science, Arlington, TX, USA), the blots were washed in 1× TBST at room temperature for 30 min. ImageJ (version 1.53q, Wayne Rasband and contributors, National Institutes of Health, Bethesda, MD, USA) was used for quantitative analysis.

### 3.13. Statistical Analysis

Results were reported in the format of mean ± standard deviation (SD). A one-way analysis of variance (ANOVA; *p* < 0.05) and a Tukey multiple comparison test were carried out to analyze the differences between the samples, using GraphPad Prism 8 (GraphPad Software LLC., San Diego, CA, USA). All data were expressed as the means ± standard deviation of three examinations.

## 4. Conclusions

In this study, SHF was extracted from *Sargassum hemiphyllum* with an average molecular weight of 1166.48 kDa (44.06%, *w*/*w*), its monosaccharide was composed of glucose (32.68%, *w*/*w*); galactose (24.81%, *w*/*w*); fucose (20.75%, *w*/*w*); xylose (6.98%, *w*/*w*); mannose (2.76%, *w*/*w*); and three uronic acids, including glucuronic acid (5.39%, *w*/*w*), mannuronic acid (1.76%), *w*/*w*, and guronuronic acid (1.76%, *w*/*w*). The SHF had characteristic FTIR spectra comparable with those of commercial fucoidans, although fucose was not its greatest monosaccharide component, unlike in other fucoidans. SHF may significantly promote NO secretion in RAW 264.7 macrophages, indicating that it has a positive immunity-boosting effect. Additionally, the SHF enhanced the immune system in RAW 264.7 macrophages via upregulation of COX-2 and iNOS levels, as demonstrated with both gene expression and protein abundance. However, the mechanisms of these immunity-enhancing effects of SHF need further research. Our results indicate that SHF promises to be a potential nutraceutical for activation of the immune system. 

## Figures and Tables

**Figure 1 marinedrugs-21-00018-f001:**
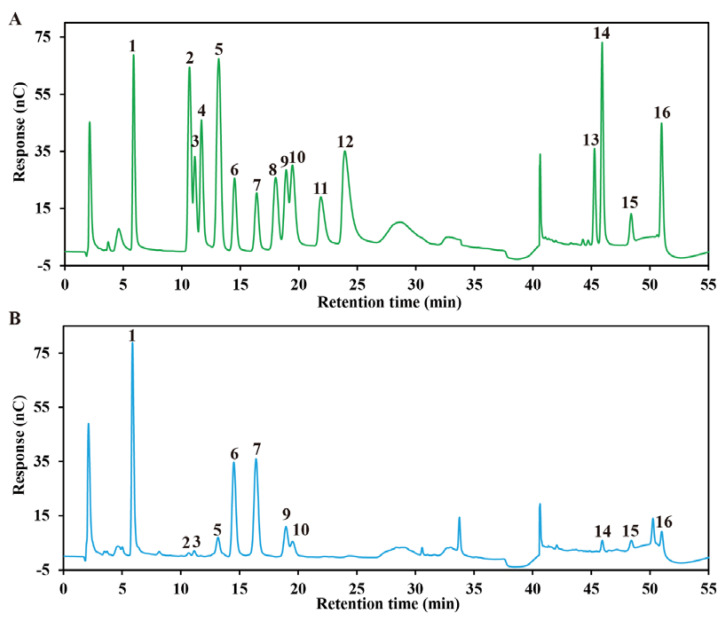
The monosaccharide chromatogram of SHF analyzed with ion chromatography. (**A**) Monosaccharide standards and (**B**) SHF. (1: Fucose; 2: galactose hydrochloride; 3: rhamnose; 4: arabinose; 5: glucosamine hydrochloride; 6: galactose; 7: glucose; 8: N-acetyl-D-glucosamine; 9: xylose; 10 mannose; 11: fructose; 12: ribose; 13: galactose acid; 14: guluronic acid; 15: glucuronic acid; 16: mannuronic acid.)

**Figure 2 marinedrugs-21-00018-f002:**
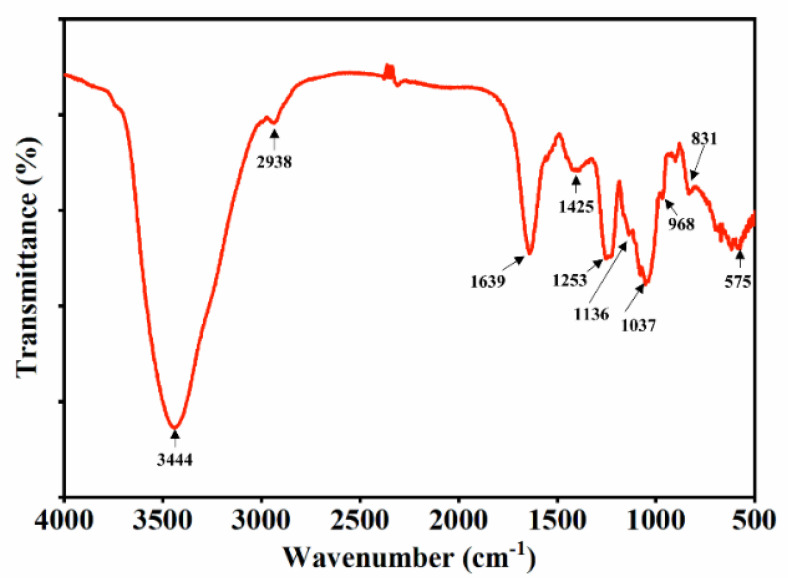
Fourier transform infrared spectroscopy (FT-IR) spectrum of SHF.

**Figure 3 marinedrugs-21-00018-f003:**
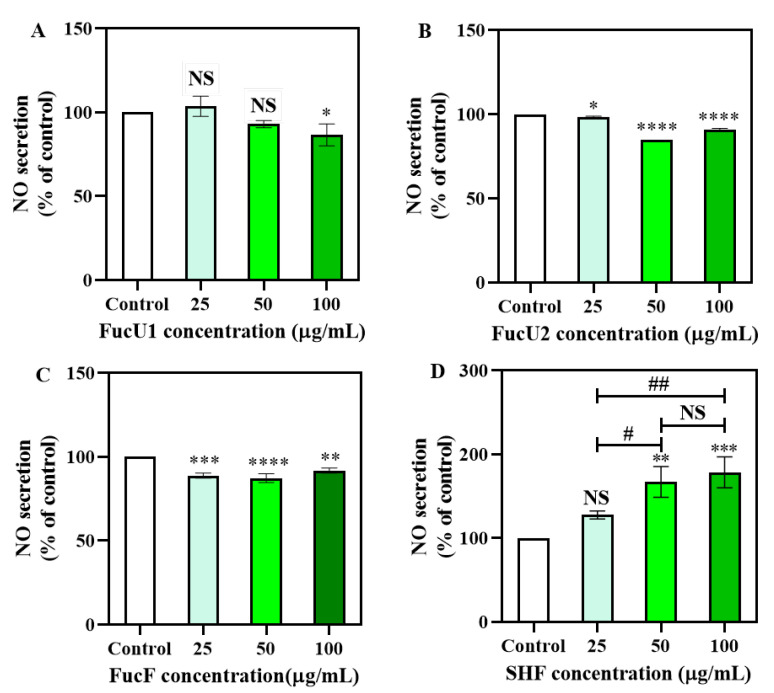
Effects of fucoidans on induction of NO production in RAW 264.7 cells. (**A**) *Undaria pitnnaifida* fucoidan (FucU1) of Qingdao Bright Moon Seaweed Group Co., Ltd.; Mw, 23.91 kDa. (**B**) *Undaria pitnnaifida* fucoidan (FucU2) of Sigma; Mw, 19.52 kDa. (**C**) *Fucus vesiculosus* fucoidan (FucF) of Sigma; Mw, 11.61 kDa. (**D**) SHF; Mw, 1166.48 kDa. NS—no significant different at *p* = 0.05; * (*p* < 0.05), ** (*p* < 0.01), *** (*p* < 0.001), and **** (*p* < 0.0001) compared with the control group, respectively. # (*p* < 0.05) and ## (*p* < 0.01) compared with each other of sample groups.

**Figure 4 marinedrugs-21-00018-f004:**
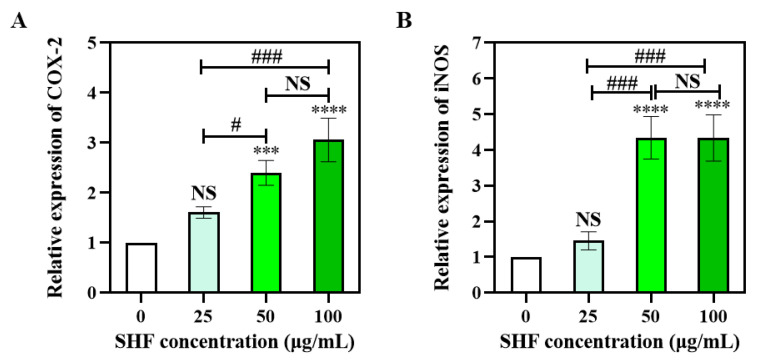
Effects of SHF on the mRNA expression levels of COX-2 (**A**) and iNOS (**B**). NS—no significant difference at *p* = 0.05; *** (*p* < 0.001), and **** (*p* < 0.0001) compared with the control group, respectively. # (*p* < 0.05) and ### (*p* < 0.001), compared with each other of sample groups.

**Figure 5 marinedrugs-21-00018-f005:**
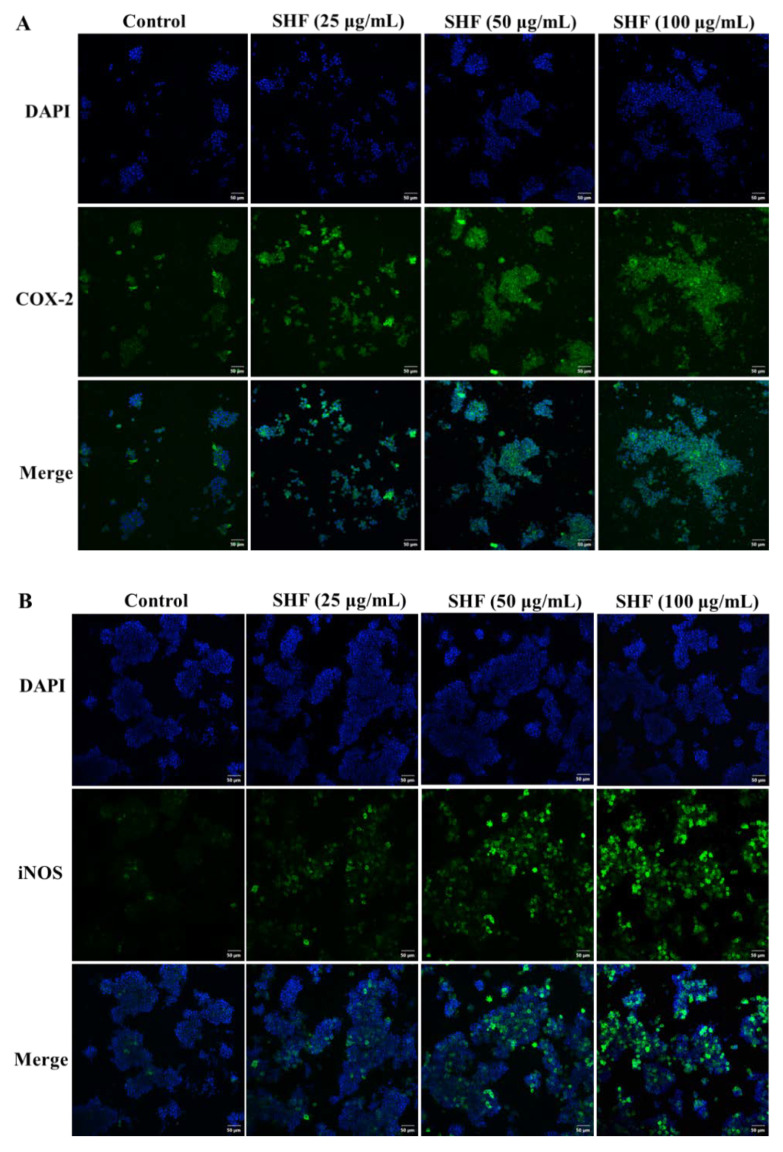
SHF upregulated the expressions of COX-2 (**A**) and iNOS (**B**) in RAW 264.7 macrophages. COX-2 and iNOS were detected via immunofluorescence staining. DAPI-stained nuclei are indicated with blue fluorescence. COX-2 and iNOS are indicated with green fluorescence. Scale bar = 50 µm.

**Figure 6 marinedrugs-21-00018-f006:**
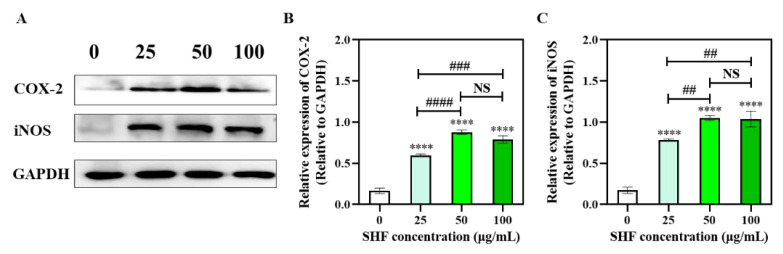
Effects of SHF (25, 50, 100 µg/mL) on COX-2 and iNOS activation in RAW 264.7 cells. Western blot images (**A**) and quantification of Western blots, COX-2 (**B**) and iNOS (**C**). NS—no significant difference at *p* = 0.05; **** (*p* < 0.0001) compared with the control group. ## (*p* < 0.01), ### (*p* < 0.001), and #### (*p* < 0.0001), compared with each other of sample groups.

**Table 1 marinedrugs-21-00018-t001:** Yields and chemical composition of SHF.

Composition	Content (%)
Yield Rate	2.72 ± 0.18
Extraction Rate	16.37 ± 1.96
Total Polysaccharide	75.35 ± 1.46
Total Protein	2.66 ± 0.67
Total Polyphenol	0.49 ± 0.01
Sulfate	44.11 ± 0.01

**Table 2 marinedrugs-21-00018-t002:** Molecular properties of SHF.

Sample	Retention Time (min)	Relative Molecular Weight (kDa)			Relative Percentage of Peak Area (%)
		Mw	Mn	Mp	
SHF	29.15	3374.86	1687.89	2097.88	7.68
	31.65	1166.48	628.33	773.35	44.06
	36.44	152.37	94.61	114.29	14.50
	36.96	121.93	76.90	92.70	13.43
	39.31	44.94	30.39	36.30	5.91
	40.95	22.40	15.90	18.87	14.42

**Table 3 marinedrugs-21-00018-t003:** The monosaccharide composition of SHF (%, *w/w*).

Monomer Category	Monomer	SHF
Neutral Monosaccharides	Glucose	32.68
	Galactose	24.81
	Fucose	20.75
	Xylose	6.98
	Mannose	2.76
	Rhamnose	1.03
	Glucosamine Hydrochloride	1.83
	Galactose Hydrochloride	0.24
Uronic Acids	Glucuronic Acid	5.39
	Mannuronic Acid	1.76
	Guronuronic Acid	1.76

## Data Availability

The data presented in this study are available on request from the first author.

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
