# Peer review of "Chemical Characterization and Immunomodulatory Activity of Fucoidan from Sargassum hemiphyllum"

_marinedrugs, 2022, doi:10.3390/md21010018_

Round 1

Reviewer 1 Report

The paper is well written and organized, however my main concern is related to the chemical characterization of fucoidan. The comprehensive analysis of chemical structure is missing. The structural analysis using NMR spectroscopy must be done. It's essential to determine the structure, hence the immunomodulatory properties observed among different polysaccharides are related to their structures. 

Reviewer 2 Report

I have read the manuscript and I have some questions and recommendations.

1. In section 2.3, please provide a chromatogram of the monosaccharide composition of SHF.

2. In Section 3.1, for FucU1, FucU2 and FucF samples of fucoidans, indicate batch numbers, date of receipt and expiration dates.

3. In section 3.2, indicate the number of the voucher and its location.

4. Previously, it was shown that the place of collection and the time of collection of brown algae greatly affect the accumulation of main compounds and, accordingly, activity (for example, https://doi.org/10.3390/md20030193). In what reproductive phase were the algae samples collected?

5. In section 3.4, please indicate the method for determining the total polysaccharide content of seaweed. Was the yield calculation done on dry weight or what?

6. For Sections 3.6 and 3.7, provide a literature reference for the method or indicate the validation characteristics of your method.

7. Fucoidan from F. vesiculosus significantly inhibits the cyclooxygenase-2 (COX-2) enzyme (IC50 4.3*10-6 g/mL) (doi: 10.3390/md18050275). Compare your data with published data. What could be the reason for the lower activity of your fucoidan compared to those previously studied?

8. Comparison of data with different fucoidans is important, but it would be more interesting to compare the results obtained with reference drugs, such as diclofenac, etc.

Round 2

Reviewer 1 Report

Thank you for your explanation, but taking into account of high impact factor of Marine Drugs, I can't accept this manuscript without a detailed NMR analysis. Comprehensive structural analysis is essential to explain the modulatory properties of fucoidan. Moreover, based on the presented data authors do not simply know whether the structure is new or already known. The purity of 75% is enough to run NMR analysis, there are some experiments that allow distinguishing the signals from two or more molecules e.g. DOSY. I strongly recommend running an NMR analysis. Moreover, in case the NMR spectra will be too complicated to solve, the Smith degradation is always a good method to obtain smaller oligosaccharides and perform the structural analysis on them.

Reviewer 2 Report

Dear Authors!

I have read revised manuscript. I have some questions.

1. How many replicates were taken per station and reproductive phase? Were the single samples big enough to limit variation (what mass and/or how many individual complete seaweeds?)
2. For the sample collection, were the samples identified fresh or dried? The corresponding sentence might be moved to further up in the description, where it was done (currently after drying and milling).

3. Previously, it was shown that the place of collection and the time of collection of brown algae greatly affect the accumulation of main compounds and, accordingly, activity (for example, https://doi.org/10.3390/md20030193). In what reproductive phase were the algae samples collected?

4. Fucoidan from F. vesiculosus significantly inhibits the cyclooxygenase-2 (COX-2) enzyme (IC50 4.3*10-6 g/mL) (doi: 10.3390/md18050275). Compare your data with published data. What could be the reason for the lower activity of your fucoidan compared to those previously studied?5. It is necessary to provide data in comparison with a standard reference compound (diclofenac, indomethacin, etc.).

Round 3

Reviewer 1 Report

Thank you for sending me the preliminary results of the NMR analysis. 

Unfortunatelly, your explanation of why you can't determine the structure of fucoidan from Sargassum Hemiphyllum is  insufficient to change my decision. 

Your explanation makes things even worse, becuase you agree that you've been working with crude, mix of different molecules, so you don't know which molecule in your isolate is resposnible for the immune-stimulating effect. I strongly recommend first purifying the isolate, then determining the immune-stimulating activity and finally determining the structure. The degradation of molecules is for the determination of structure, not for immune-stimulating studies. 

In my opinion, based on the spectra sent to me,  ithe structure can be partially determined (there are at least 2-3 fucose). However, the quality of the spectra should be better, maybe you should increase the number of scans or test more amount of the material. 

Reviewer 2 Report

I have read the revised manuscript. I have no more questions.